# Forced Handling Decreases Emotionality but Does Not Improve Young Horses’ Responses toward Humans and their Adaptability to Stress

**DOI:** 10.3390/ani14050784

**Published:** 2024-03-02

**Authors:** Inês Pereira-Figueiredo, Ilda Rosa, Consuelo Sancho Sanchez

**Affiliations:** 1Neuroscience Institute of Castilla y León, University of Salamanca, C. Pintor Fernando Gallego, 1, 37007 Salamanca, Spain; sanchoc@usal.es; 2Thekidsfellows-Research Group in Anthrozoology, 6060-309 Idanha-a-Nova, Portugal; thekidsfellows@gmail.com; 3Animal Behaviour and Welfare Laboratory, Center of Interdisciplinary Investigation in Animal Health, Faculty of Veterinary Medicine, University of Lisbon, 1300-477 Lisbon, Portugal; 4Department of Physiology and Pharmacology, University of Salamanca, 37007 Salamanca, Spain

**Keywords:** animal welfare, human–horse relationship, early handling, stressful conditions, forced handling

## Abstract

**Simple Summary:**

Human and horse interactions, according to the literature, are important to animals’ welfare and health. In a previous study by our group, early handing in foals proved to decrease fearfulness and improve manageability and behavioral responses toward humans. In the present study, we investigated the influence of two different types of short-term handling procedures: one more intensive approach (daily touch and rubbing), compared to handling monthly, and one leaving foals undisturbed. Right after weaning, we measured the behavioral and physiological responses of all of the foals to new stressful environments, veterinary examen procedures, and unknown persons. Intensive or monthly handling had the same effects on novel stressful situations. Handling young horses, regardless of the timeframe, improved responses to stressful situations and environments. However, handled foals did not seek contact with humans and had an increased neutrophil–lymphocyte ratio profile. We expected the handled foals to be more prompt to face new stressors. However, handling associated with no control of the humans’ actions may not be viewed positively by the young horses. Our results reinforce the need to use physiological parameters, together with behavioral observations, to monitor stress responsiveness and welfare.

**Abstract:**

Horses are often still exposed to stressful or inadequate conditions and difficult relationships with humans, despite growing concerns about animal welfare. In the present study, we investigated the impact of different approaches of short-term handling sessions on young Lusitanian horses raised on a high-breed farm, specifically on their later adaptability to humans and stressful environments. Thirty-one foals (3 months old ± 15 days), from both sexes, were separated into three groups, one submitted to 3 consecutive days of handling sessions (Int-H), another to one handling session each month for 3 months (Month-H), and one left undisturbed (control). At 8 months old ± 15 days, all foals were evaluated during behavioral tests (restraint in a stock and forced-person test). Evaluations were based on behavioral observations and physiological assessments. The handled foals (Int-H and Month-H) reacted less to being isolated and restrained and better tolerated human contact and veterinary procedures than the control ones. The handled foals displayed less evasive and negative behaviors toward human approach, but also sought less human contact and did not interact, regardless of the handling timeframe. All animals displayed signs of stress when restrained in the stock, with increased neutrophil counts and CHCM levels in the blood, and no differences in metabolic (CK and LDH) and other hematological parameters. The neutrophil–lymphocyte ratio was significantly higher (*p* < 0.05) in handled foals than in control ones, suggesting low standards of welfare. Our data suggest that early forced handling decreases fearfulness in new environments; however, it does not improve the horses’ relationship with humans, and it decreases welfare.

## 1. Introduction

Humans and horses, for the last few centuries, have shared a common history, having a close relationship and establishing strong bonds [1,2]. But horses have not always been fully understood, and they have not always understood human activities. This paradigm can be explained by considering equines’ evolution as a prey animal and remembering that their domestication is quite recent.

Horses have been evolving for millions of years as sensitive animals, very perceptive of their surrounding world [3]. In modern societies, the social and environmental conditions in which domestic horses are reared often are quite different from those in which they evolved as a species [3,4,5]. Sometimes, horses perceive their environment as stressful and dangerous, even if humans are not aware of it. Frequently submitted to social deprivation [6], contained in individual housing [7,8], tallying difficult interactions with humans and training systems [9], and subjected to loud noises, invasive procedures [10], frequent transportations [11], and consequent changes in social or physical environments, horses may become easily frightened and stressed.

Any of the described conditions is of huge importance to horses’ welfare and may cause serious health problems [4] that can shorten their working life and bring economic costs to their trainers or owners. This may result in immunosuppression [12] or learning issues [5,11], including a lack of motivation [9,13], and can be the cause of many accidents, mostly because of highly disproportional emotional responses.

Emotional behavior, reactivity, or emotionality is defined as the “set of behavioral and physiological responses that occur in anxiety producing circumstances” [14], and determining how to control it is a main factor in the equine industry. Identifying any strategy that would help to reduce the fearfulness of horses and modulate their behavioral responses to situations and humans, particularly when facing potentially stressful conditions, is an essential component impacting not only their health but also horse-related incidents and accidents with their handlers.

Behavioral and physiological responses to stressors are clearly influenced by social interactions, including maternal interactions [15]. In the south of Portugal and Spain, it is very common for young horses to be reared on large breeding farms and kept together in big groups of the same age on pastures. These social conditions are potentially positive for equines when compared to individual housing conditions [16,17,18]. Horses kept in groups learn quicker, develop fewer undesirable behaviors, and require less time for desensitization to novel equipment [18,19]. Nevertheless, on such high-breeding farms, fillies and colts are “artificially” put together, through the humans’ decision, which is very different from the natural composition of horse social groups [6]. Furthermore, on these breeding farms, the interactions of the young animals with humans are normally based on short occasional veterinary inspections, and apart from feeding, positive interactions are very rare. Altogether, horses kept in such breeding conditions can develop potentially difficult relationships with peers and humans.

From this perspective, recent studies converge on the idea that promoting positive interactions with humans can be a way to improve horses’ emotionality, social bonds, and well-being [20,21,22,23,24,25]. It is argued that human company may be important to a horse, where physical contact and the benefits of stroking are crucial elements for potentiating human attachment [23,24,25,26]. Likewise, the frequency and number of interactions can determine how horses deal with situations curated by humans. A review of the literature suggests that human interactions may simply consist of habituation to any contact stimulus (brushing, haltering, etc.) [21,22,25], and the continuous repetition of such stimuli, until they become neutral, has been named handling [26,27,28].

Early handling of young horses is suggested to be a good strategy for improving relationships with humans [28,29] and manageability [30] and reducing emotional reactivity or fearfulness [31,32,33]. However, some studies have led to contradictory results, possibly due to the heterogeneity of study designs, the time handling was performed, and the type or duration of the contact. Furthermore, research on this topic quite often uses horses from different breeds and different housing conditions, without knowledge of previous contacts. Regarding the high influence of individual differences on temperament traits and behavioral responses toward humans [34,35,36,37], controlling such variables can be of utmost importance. To our knowledge, our study is the first one testing behavioral responses on young horses of the same age and breed raised under the same conditions, and this can provide a great opportunity for a better understanding of human interactions and handling practices during the development of horses.

Currently, there is a growing interest in biomarkers of good or poor welfare. The most common physiological indicators of stress and discomfort, in several species, include cortisol levels (blood and saliva), muscle tension, and heart rate (HR) variability [20,38]. Nevertheless, until now, there is still a low conformity in the reliability and validity of such measures of horses’ emotional states. The HR response to experimental situations varies a lot among individuals, and HR increases with emotional reactivity are not always displayed behaviorally [19]. Cortisol is often not the best indicator of stress and of poor welfare in horses, as its secretion increases with physical activity, has a daily fluctuation [39], is influenced by gender, and may decrease in horses that experience welfare changes [17].

Although there is an inherent connection between behavioral and physiological responses to stress, it has not been thoroughly studied. Ethograms usually comprise objective and instantaneous indicators of emotional behaviors and welfare status in animals [16,40], but it is still difficult to the measure emotional states of horses [21], and most of them are only revealed under high-anxiety conditions [31]. In a previous study by our lab, it was shown that hematological (hematocrit, platelet, and leukocyte counts) and biochemical parameters (protein and cytoplasmic enzyme total counts) were altered in rodents exposed to prolonged stress, even months after it had ended [41], suggesting that those markers could be good indicators of chronic stress. Also, better than leukocyte counts, the neutrophil–lymphocyte ratio (NL ratio) was suggested as a reliable biomarker for measuring stress in vertebrates [42]. And recently, some authors proposed the assessment of the NL ratio as a good indicator of the stress and welfare status of horses [16,43,44].

In the present study, it was hypothesized that handling interventions, early in life, could be a positive experience for horses, reducing fearfulness and improving resilience and emotionality upon exposure to unpleasant and new stressful situations. By comparing outcomes in different approaches of handling procedures (more or less intense), the present study hopes to contribute to a better understanding of the impact that human activities have on the development of later behaviors, welfare, and quality of life of equines.

## 2. Materials and Methods

### 2.1. Animals and Management

Considering the genetic component linked to emotionality and temperament traits, we only evaluated Lusitanian foals [43]. The National Stud Farm where they were born is located in Southern Portugal. As a usual practice of the Stud Farm, until weaning, mares and foals (a total of 57 born in the end of winter and spring of 2021) were kept on the pasture during the night (from 4 p.m. until 8 a.m.), being housed every morning. Before the experiment, the foals were not handled except for a brief physical examination when they were only a few hours old. Those selected to take part in the present study (N = 31) were identified at birth and allocated to one of the three groups (see Figure 1), considering their partition as equally possible by age (only those born from March to April), sex (16 fillies and 15 colts), and father. Starting at 3 months ± 7 days old, two of the groups of foals were submitted to short-term handling sessions (each one lasting 10 ± 5 min). One group was submitted to daily sessions (N = 12, intensive handling: Int-H), another to once-monthly sessions (at 3, 4, and 5 months old: N = 9, monthly handling: Month-H), and the last group was left undisturbed (N = 10, control).

All handlers were experienced students (all males) from the Equine College of Alter-do-Chão with at least two years of experience in riding and managing horses. During the study, the foals were reared as usual, being in close contact with each other, before and after weaning (at approximately 7 months old ± 1 month), when they were kept together in a large (16 × 20 m) outdoor paddock until the end of the trial.

The experiments were conducted on the same stud farm where the animals were born, and we confirm that the National Stud Farm, as the owner, provided informed consent for the present study. The authors confirm that animal welfare was never at risk and that the experiments complied with the policy relating to animal ethics and welfare (Article 1, number 5, Directive 2010/63/EU of the European Parliament and of the Council of 22 September 2010 on the protection of animals used for scientific purposes, and Article 2, number 7, of the Portuguese Legislation number 113/2013 of August 2013).

### 2.2. Handling Procedure

The handling procedure was adapted from our previous work [31] and followed the methodology first described by Miller [44] and modified by Henry [28] and Schmideck [29] regarding the time it was performed and it being performed with the foals standing up. Three months of age was the timeframe chosen for the start of the experiment, as this is the usual age for the first consult performed by a veterinarian for routine examination. In the present study, the handling contact was forced, consisting of individual contact performed by two experienced handlers, starting with restraint, haltering, and touching the whole body with empty hands, which included invasive procedures (touching inside the mouth and nostrils). It was performed as gently as possible (the handlers controlled each foal by using subtle pressure cues and promptly releasing pressure and stopping if signs of displeasure were shown) and for a short time. Some of the desensitization actions we previously used [31] were excluded, such as the use of a plastic bag and tapping on all four feet, and the procedure was focused more on socialization and causing the foal to relax. All handlers were required to have previous experience with horses, and during the procedure, they were required to have a good attitude and a calm emotional state and to use a quiet and pleasant voice.

During handling sessions, each foal remained in the usual environment for the day—an open pen (100 m × 7 m) where they were routinely fed—with the other mares and foals around (about 60 females, at more than 3.5 m distance) and with its mother standing close. One handler (H3) held the foal’s mother while she was eating and brushed her, as previously suggested [29], or kindly talked to her to calm her down, trying not to affect her behavior toward her foal [33]. The first handler (H1) remained still for about 1 min and then slowly started the approach to the chosen foal, diagonally to the animal’s shoulder, staying on its visual field, making gentle sounds but avoiding direct eye contact, as previously described [31]. When near the foal, H1 gently restrained it by placing one arm around its chest and the other one over its hock, hugging it, and then attempted to put the halter on. The second handler (H2) only approached and helped in restraining if the foal tried to step away (this procedure took ± 3 min). Once the foal was restrained, H1 gently started tapping and brushing the animal’s body, gradually from the head to the legs and all over the body, except the belly. The intensity of brushing was increased, but brushing was stopped if the foal did not relax (for another 4 min). The tactile stimulus continued, and then H1 attempted to interact with and, in the last 2 min of the session, tried to calm the foal, in closeness (neck and shoulder in contact with H1).

### 2.3. Behavioral Tests

One month after weaning, at 8 ± 2 months old, all foals were assembled (N = 57) in an outdoor paddock with an old mare, and those selected for the present experiment (N = 31) were subjected to two consecutive days of behavioral tests, in unknown environments and with unknown humans.

#### 2.3.1. Stock Test

On the 1st day of the behavioral tests, starting at 10:30 a.m., the foals were freely grouped 5 by 5 and taken from their outdoor paddock (Figure 2A) to be led into a stock (1 × 2 m) (Figure 2B), where they were individually restrained, identified, and submitted to veterinary routine procedures. This was the first time that each foal was restrained and isolated from its conspecifics. While inside the stock, if belonging to any experimental group, each foal was subjected to a behavioral assessment, once entering, and after a pause of some seconds, during ±4 min. Afterward, the foal was let out from the top of the stock.

The response to each procedure was scored, and the scoring included evaluating the behavioral reactions to (a) being isolated and restrained (restraint), (b) being touched in the neck and back and haltering (contact + halter), and (c) blood sampling and deworming (Table 1). For data analysis, each foal was evaluated using behavioral scores (0–5 points), and the frequency of behaviors (number of times) was assessed using a method adapted from our previous study [17].

Vaccination, deworming, and blood sample collection were performed with the foal inside the stock, immobilized from outside, by one equine veterinarian (in ignorance of each foal’s status) and two unknown handlers holding the foal carefully, but firmly if needed, with a rope. Blood samples (±10 mL) were collected by jugular venipuncture, from each foal, into two different tubes; the samples were preserved on ice while the behavioral test was being performed and later transported to the Stud Farm’s laboratory to be analyzed. Two foals from the initial group were excluded from the study analyses because they did not allow blood sampling after 12 min of attempts. Fecal samples were collected from the stock floor (5–10 g).

#### 2.3.2. Forced-Person Test

On the 2nd day of testing, all foals were evaluated in an unfamiliar arena (5 × 6 m pen), with the walls covered by a 1.7 m high dark net (Figure 2C). During the first part of the test, each foal was isolated and forced to stay in the presence of an unfamiliar person who was already inside the pen and remained still for 60 s (motionless human test (MHT)) [31]. The latency to entering the zone (Lat zone) was assessed, and the evaluation of the individual reactions to the motionless person, while the animal was isolated (Rx isolation), was adapted from the previous trial described in our study [31] (Table 1).

In the second part of the test, the handler was allowed to approach the foal and attempt contact for another 180 s (approach–contact test). The approach style was indirect (relaxed, no rope swinging and no direct eye contact), as suggested by Birke [34], and the contact was not forced. For each foal, the distance of approach that was allowed without fleeing (D Apo (in m)) and the latency of time to be touched (Lat touch: in s) were assessed by one person uninformed of each animal’s status (O1). The approach was not stopped if the foal moved by walking, and if its neck was within 1 m distance, the handler could attempt contact.

The responses to the human’s presence inside the pen and contact were scored (Table 1) as the quality of contact and evasive behaviors (attempt to retreat by walking away, trot, canter) and were rated by two different persons, previously trained (O2 and O3), located on each side of the arena (intra-observer variability above 80%). The test was video-recorded using a camera (Sony HDR-CX 220) mounted on the top of one wall (opposite the stock side). The occurrence of positive (investigatory) and negative behaviors (defensive or any direct attempt to threaten the person) (Table 2) was evaluated by the total behavior duration shown by each animal, throughout the duration of the test (time in s), and was later analyzed in detail by the same trained researchers.

### 2.4. Hematological and Biochemical Analysis

Blood samples were collected in two different tubes during restraining in the stock test. For the hematological analyses, blood samples were collected in EDTA (K3)-containing tubes that were freshly processed using an automatic cell counter (ADVIA 120 cytometer, Bayer, Leverkusen, Germany) in the Stud Farm’s Clinical Analysis Laboratory. The parameters evaluated were erythrocyte counts, hemoglobin concentrations, mean corpuscular hemoglobin concentrations (MCHCs), hematocrit values, platelet counts, mean platelet volume (MPV), and leukocyte cell counts (WBCs) and type.

For the biochemical analyses, blood samples were collected into heparinized tubes (Monovette Li-Heparin, SARSTEDT AG & Co., Numbrecht, Germany) that remained at room temperature until being centrifuged at 10,000× *g* for 20 min to obtain serum, which was drawn into Eppendorf tubes and used fresh on a SPOTCHEM^TM^ EZ device (Woodley, Lancashire, UK). The levels of serum total protein (T-Pro) and the cytosolic enzymes glutamic-oxaloacetic transaminase (GOT), creatine kinase (CK), and lactate dehydrogenase (LDH) were measured using the Multi-Parameter Strips Liver-1 ARKRAY^®^, according to the manufacturer’s directives.

### 2.5. Parasitological Analysis

Fecal samples were collected from the ground of the testing settings, put in a plastic bag, stored at 4 °C in a cooler with ice packs, and examined within 48 h of the collection in the laboratory. A modified Wisconsin double-centrifugal sugar flotation technique was used for all fecal egg counts (FECs) as previously described [22], with a minimum detection limit of 1 egg/g. Slides were examined at 10× magnification under a standard light microscope. Eggs were counted and classified according to their morphology.

### 2.6. Statistics

Statistical analyses were performed using IBM ^®^ SPSS^®^ software, version 20 (IBM Crp. and SPSS Inc., Chicago, IL, USA, 2011). During all experiments, the foals’ behaviors were recorded continuously. The analyses were carried out using nonparametric or parametric statistical tests, according to the variable characteristics, with a significance threshold at 0.05. For each behavioral test and each behavioral item, the independent samples (experimental groups) were compared by using the Kruskal–Wallis (to analyze nonparametric variables) or one-way ANOVA tests (to analyze parametric variables), followed by the Fisher PLSD test for post hoc comparison. Then, a rank was assigned to each foal and each behavioral item. The sums of the ranks were compared using the Kruskal–Wallis test, and to assess the degree of correspondence among scores, Spearman rank correlation coefficients were determined. The value of minimal significance was considered at *p* < 0.05.

## 3. Results

### 3.1. Behavioral Tests

Our aim with the behavioral tests was to determine the handling effect in young horses exposed to stressful procedures, such as isolation in new environments, and to human presence, contact, and veterinary procedures. Handling significantly affected the foals’ responses inside a stock (Figure 2B), with the handled ones displaying lower reactivity when isolated and restrained, regardless of the intensity of handling (daily or monthly) (χ^2^(2) = 13.6, *p* = 0.03). Kruskal–Wallis H tests also revealed statistical differences in foals’ reactions to human contact and to haltering (χ^2^(2) = 12.7, *p* = 0.042), and when compared by pairs, we found that Int-H foals tolerated human contact better than controls (0.95 vs. 2.08, *p* = 0.031). The administration of vaccines and paste dewormer and blood sampling were not affected by previous handling (χ^2^(2) = 4.1, ns), and no differences were found among experimental groups during veterinary procedures inside the stock (see Figure 3).

On the second day of testing, during the forced-person test (Figure 2C), in the presence of a stationary person (MHT), our results showed that handling did not affect the behaviors when foals were isolated (Rx isolation) or the will to approach (Lat zone) (see Table 3). The ANOVA tests revealed no effect of group on the distance foals allowed the handler to approach without flight (D Apo, ns) or the time taken to allow contact (Lat touch, ns) (F_1,31_ = 1.92 and 2.7, respectively). Nevertheless, differences were found in mean scores for evasive responses (*p* = 0.015) and the total time the negative behaviors (*p* = 0.028) were exhibited, F_1,31_ = 4.8 (Table 3). Post hoc analyses showed control foals were more difficult to reach and displayed more defensive reactions (*p* < 0.05) toward the unknown handler, when compared to handled ones, regardless of the handling procedure. The quality of contact was significantly improved as an effect of handling (χ^2^(2) = 11.5, *p* = 0.031), and post hoc analyses revealed that Int-H foals allowed more body areas to be touched than control ones (*p* = 0.022). When the total time exhibiting positive behaviors toward an unknown person was assessed, it was determined that handling also affected the will to positively interact with the handler (F_1,31_ = 12.8 *p* = 0.039), and post hoc analyses showed that Int-H foals exhibited significantly fewer positive interactions than control ones (9.5 vs. 30.2, *p* = 0.028), with the same trend in monthly handled ones (*p* = 0.052).

The nonparametric measures during behavioral tests revealed a strong correlation between the scores for behaviors in both environments while the foals were isolated (rs = 0.68, *p* < 0.001) and a significant correlation in response to unknown environments and unknown humans (veterinary procedures vs. D Apo: rs = 0.57, *p* < 0.01; veterinary procedures vs. quality of contact: rs = −0.37, *p* < 0.005), and no effects affecting the behaviors of foals were correlated with their gender (males/females).

### 3.2. Hematological Analysis

As shown in Table 4, the procedure of restraining foals affected the neutrophil counts in peripheral blood in all animals. The WBCs and lymphocyte profiles were not affected, and the cytosolic enzymes CK and LDH showed normal values with no differences among groups (*p* > 0.05) for any parameters, except for the neutrophil–lymphocyte Ratio. Significant differences in the NL ratio were found as an effect of handling (χ^2^(2) = 21.5, *p* = 0.047). The NL ratio levels of previously handled foals were over the reference values, and post hoc analyses revealed that Int-H foals presented higher levels than control ones (2.55 vs. 1.46, *p* = 0.042), with no differences in Month-H foals.

### 3.3. Body Weight, Parasitological Analysis, and Health Status

Assessment of body weights and analyses of shedding categories in feces (eggs/g) on the days when behavioral tests were performed showed that 100% of foals were at normal weight for 8-month-old Lusitanian fillies and colts (130 ± 15 kg and 150 ± 20 kg, respectively), and mean levels of intestinal egg counts (*Strongyle*, *Parascaris* spp., and *Oxyuris equi*) were under normal values (<500 epg) [45].

## 4. Discussion

The present study aimed to assess if short-term handling experiences in young horses would reduce fearfulness and improve later responses toward humans and to new and stressful situations. To this end, foals submitted to different intensities of handling procedures (daily or monthly sessions) were compared with one undisturbed group. According to previous research, the results obtained showed that three sessions of early handling, regardless of intensity, resulted in learned behavior, with decreased self-defense and emotional responses toward humans and stressful situations [25,26,30,31]. However, in the present study, it was also perceived that handled foals freely in the presence of an unknown person did not seek human contact, had less interest in interacting than undisturbed controls, and exhibited signs of prolonged stress. It was found that hematological parameters, along with behavioral observations, were useful for monitoring stress responsiveness.

The present study took into account concerns for welfare, and so caution was taken while handling to reduce any unnecessary stress.

### 4.1. Handling Decreased Emotional Reactivity, Regardless of Its Intensity

Handling, stroking, or “gentling” [29] an animal is traditionally assumed to be a positive procedure and to have beneficial effects on animal behavior and physiology of some species. But in horses, while some studies state that forcing human contact or handling is perceived as positive [25,28,29,30], others disagree [33,34,36], making the topic controversial. With handling, fundamentally, the animal learns through habituation [27] to identify irrelevant specific stimuli and diminishes reactivity toward them [12]. This is a feature of extreme importance in equines due to equines being large and highly reactive animals in which sudden defensive responses to pain or fear may easily injure handlers, veterinarians, or trainers [39,46].

In the present study, it was observed that early handling decreased later reactivity or emotional responses of the young horses to being restrained in a stock and touched and haltered by one unknown person, with no differences in response to blood sampling and deworming. None of the handled foals achieved the highest reactivity response (score 5) during the routine veterinary procedure or harmed themselves (compared to three from the control group). Notwithstanding, as demonstrated in this study, more information is needed regarding the emotional states of the horses, whether positive or altered, and the horses’ responses toward humans.

### 4.2. Handling Decreased Negative but Also Positive Behaviors toward Humans

When horses face a potential danger, the first behaviors they exhibit are usually flight or fight, eventually followed by an explorative behavior. In line with this, it was observed that all the young horses perceived the behavioral tests as stressful situations, exhibiting evident behavioral and physiological responses.

In the present study, the interactions of a free foal toward an unknown person were examined. Horses naturally seek human attention when they are alone [37], but the data herein showed that no foal spontaneously approached the person standing nearby (during the MHT). These results also showed that, as expected, during approach–contact tests and when physically alone, control foals were the animals that showed more escape responses to a person approaching in the unknown paddock [31], were more difficult to reach, and vocalized and chewed more. But control foals also explored the unknown environment and the unknown person more. It was observed that previously handled foals moved slower and carried their head lower, signs showing that they were apparently less emotional [47].

During the forced-person test, the young animals were allowed to move in a small enclosure, so they had some options regarding their behavioral responses to the person. In accordance with our previous study [31], the handled foals allowed the person to move closer and have better contact, but control foals were the only ones that sought the attention of the person and explored, smelled, and were curious about the person; these behaviors are associated with a positive mental state [23].

The person inside the paddock was unknown to the foals, but it is not certain if the presence of a familiar person would help the young horses feel safer while facing potential threats [24], and they would most probably generalize their responses [28]. For example, Schmidek [28] observed that gently handled foals did not discriminate between handlers regarding their familiarity or experience, while Ijichi [24] observed the opposite when comparing horse owners to unfamiliar handlers during mildly stressful handling procedures. However, more research is needed on the extent of generalization of handling experiences.

In the present study, the handling procedure was very brief and occurred in a known environment for the foals (the pen), with their mothers and other foals around, and that may have prevented their responses to humans. Horses do not find humans more prominent than conspecifics as companions [3], suggesting that attachment to people is more likely to manifest in the absence of conspecifics, which represents a significant welfare issue and may explain the impact of handling at the specific timeframe of weaning that was previously detected by the authors of the present study [31].

Human caregivers feel strong bonds toward their animal companions. Horses also prefer individuals with whom they had positive experiences, but unlike dogs, horses are not dependent upon their familiar caregivers; they do not feel separation distress or safety issues toward someone known [36], suggesting that the handler’s competence and attitude, while interacting, are greater factors that may impact horses’ behavior and trust [21,23].

According to some authors [20,21,22,23,24], human–horse attachment is highly desirable for achieving good welfare in both species, but for horses, bonding to humans is apparently highly complex [23]. The present study showed that early handling of the foals decreased not only the negative (e.g., flight responses) but also the positive behaviors (e.g., attention or approach will) exhibited toward the person present. Our results may suggest that handling, daily or monthly, decreased emotional responses toward humans but did not improve bonding will.

### 4.3. Handling Affected the Response to Stress and Foals’ Welfare

For horses, being isolated and facing new, unknown challenges is stressful [10,37,48]. In our study, we detected a general increase in neutrophil counts in response to acute aversive experiences, such as the circumstances of being restrained and submitted to veterinary procedures inside a stock. Those changes, resulting from the big increase in glucocorticoid hormones, are a normal body response in mammals [40,49], having notorious advantages for an animal’s short-term survival.

Some authors suggested that early experiences may be determinant of the ability of animals facing stress by modulating the nervous system and, eventually, leaving their marks throughout life [32,33]. Thus, we expected the handled foals to be more prompt to face new stressors. However, we found no significant changes in erythrocyte, leukocyte, lymphocyte, or biochemical counts, and on the other hand, Int-H foals exhibited significantly higher NL ratios than control ones.

A significant increase in neutrophil counts in peripheral blood has been reported in several species just after 6 min of a stressful event, exercise, or inflammation of the muscle tissues [50]. In this study, it could be demonstrated that cytosolic enzymes in the blood (e.g., CK and LDH parameters) were not altered in any experimental group, meaning that the neutrophilia was not linked to muscle injury or inflammatory state and most probably was linked to the stress condition. 

Increased neutrophil counts accompanied by lymphopenia are becoming a recognized biomarker of prolonged stress in animals, for a growing number of researchers [42,51,52]. Our results agree with previous works, e.g., [7], asserting even more the importance of the indicative value of this ratio (NL). In a recent study, Popescu [16] found increased values of the NL ratio in working horses and suggested that these values were probably a consequence of inadequate housing and management after investigating the relationship between the NL ratio and welfare scores.

In the present study, the handled foals showed mean values of the NL ratio just a trace above reference values, meaning that there were some individuals for which the NL ratio was higher. Moreover, comparing the range of NL ratios, we observed that 70% of control foals were within the normal range for healthy horses (1.25~1.57) [52], but only 10% (Int-H) and 12% (Month-H) of the handled ones were also within this range, and all the others exhibited higher values.

We cannot conclude that our findings altogether indicate a health problem in the young horses, as we could establish that all other parameters were at normal values (after examining weights, biochemical parameters, and parasitological status), and our findings cannot be assigned to issues in the working program, housing, or management, as all foals were reared together in the same conditions. This infers that the elevated NL ratios we found rather suggest discomfort and poor welfare in previously handled foals.

The pattern of handling used in the present study was not likely consistent with the foals’ natural needs, particularly those related to interactive behaviors. The tactile human contact was forced, and the foals were physically restrained by unfamiliar handlers using unusual sounds and smells, leaving no sense of control; the contact was most probably perceived as invasive and stressful and affected later responses to stress and welfare. Moreover, it is currently well established that early adverse experiences cause changes in the nervous system that persist throughout life [53]. This suggests that handling experiences in young horses, even when ending up to 3 or 5 months before subsequent assessment, may have had a high impact in regard to the future susceptibility to stress and the individual’s reactivity. However, more research is needed for a better understanding of the relationship among handling practices, psychological stress, and long-term neutrophil function in equines.

### 4.4. Limitations of the Study and Concerns about Handling Practices

When Ligout and her colleagues [30] compared different methods of handling, “forced” vs. “unforced” contact, they found more improvements in the foals’ relationships with humans in forced-handled ones. In their study, these foals approached and stayed closer to handlers, allowing the handlers to touch them longer than the control and unforced ones, which did not approach the handlers at all. In our study, during handling, the foals had no control over their own behavioral choices, and they exhibited less fearfulness and decreased reactivity toward humans and environments when tested months after.

Nonetheless, animals that are exposed to inescapable conditions may stop responding to a stimulus, first by habituation, but afterward, they may develop some learned helplessness feelings [31,53,54,55]. Learned helplessness reflects the despair felt when there is no control over circumstances, and it has been proposed as a model of the “stress and coping” paradigm [26]. Horses that are unable to avoid contact, or for which flight responses are physically prevented through restraint, may be subjected to some degree of flooding and, potentially, learned helplessness [54]. Some authors (e.g., [6]) suggest that teaching horses while giving them the possibility of retaining a certain degree of “choice” over human demands can improve welfare and that these animals will associate human actions with positive experiences.

Overall, our results must be taken cautiously as they have some limitations. The non-independence of the study participants may have affected the groups’ behavior and hematologic findings—future research testing the external validity of these findings with an external group of horses is needed. Other important limitations arise from the small sample sizes and the specificity of handling procedures used in our study, as we recognize that small sample sizes can increase the risk of false positive findings. The present study is part of a big project that aims to test the effects of different timeframes of handling on the later responses of foals; therefore, it did not consider a wide range of handling methods or other potential factors that could influence the outcomes, which may limit the generalization of the results.

Furthermore, some correlations between health-related welfare indicators and behavioral responses of the assessed horses as follow-ups would be helpful. Still, it must be noted that it is of utmost importance to consider the patterns of behavioral and hematological changes described here in further research.

## 5. Conclusions and Recommendations for Future Research

Our data reinforced the idea that handling associated with forced restraint and no control of humans’ actions may not be seen by young horses as a positive event. Horses subjected to short-term handling sessions, regardless of their timeframe, exhibited improved acceptance of new stressful situations, with fewer defensive or aggressive behaviors, compared to non-handled ones. However, the handled foals showed less interest in humans, exhibiting signs of negative states and signs of distress.

Simply caring for animals is becoming not enough. Horses with altered welfare are more prone to negatively perceive situations, which is a feature of utmost importance for the safety and health of themselves and the people around them. To date, limited research has explored the impacts of handling and human interactions on horses’ health and welfare, focusing mostly on the removal of emotionality. Nevertheless, it is highly recommended that research about handling procedures that promote positive horse–human relationships include the assessment of positive emotions, together with the positive perceptions of animals about their environment and the people around them. Welfare commitment is needed, in order to define the more favorable situations for animals and to promote the best practices. Finally, while raising, handling, training, or preparing horses, we need to understand the way they perceive their environment, the way human actions are assessed, and that these perceptions are different from ours. The present study shows that for horses, a lack of choice about their surroundings is perceived as stressful and greatly affects their mental state and quality of life.

## Figures and Tables

**Figure 1 animals-14-00784-f001:**
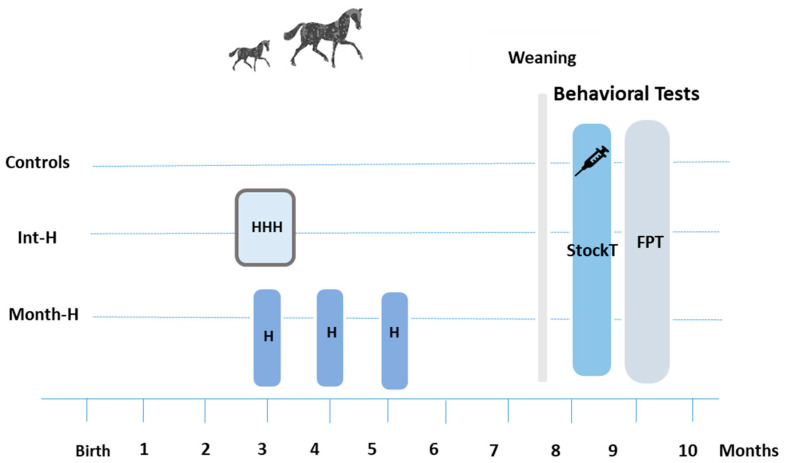
Experimental design used in the present study. Beginning at 3 months ± 7 days old, foals were subjected to forced handling daily for three days, Int-H (N = 12), or once monthly, Month-H (N = 9), while controls (N = 10) were left undisturbed. After weaning, all experimental foals (N = 31) were subjected to two consecutive days of behavioral tests, including isolation, restraint, blood sampling, and deworming. Abbreviations: Int-H: intensively handled group; Month-H: monthly handled group; StockT: stock test; FPT: forced-person test; H: 1 session of handling procedure; HHH: 3 sessions of handling procedure.

**Figure 2 animals-14-00784-f002:**
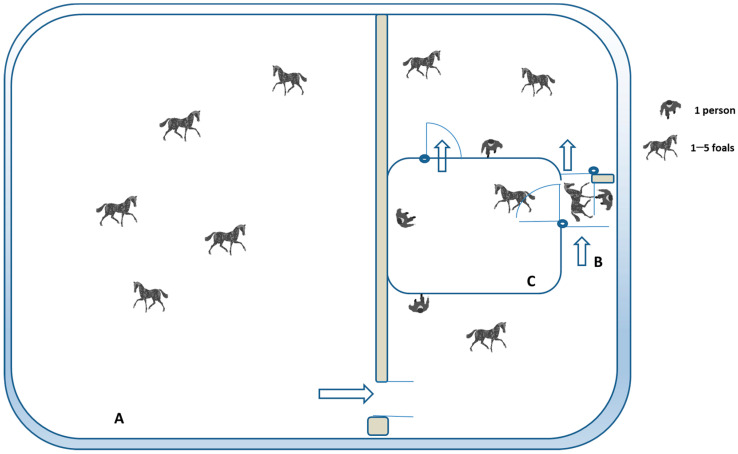
Overhead view of the testing pen and the outdoor paddock used for the behavioral tests. All foals (N = 57) were in a 16 × 20 m outdoor paddock together (**A**). In the days of testing, the foals were led freely to a stock (**B**) to be tested during restraint in the stock test or to the open arena (**C**) for forced-person test evaluations. When being tested, each animal was surrounded by its peers and by hidden observers on all sides. A dark net did not allow visual contact with other foals and separated them physically. The persons present were 2 researchers, 2 veterinarians, and 3 handlers.

**Figure 3 animals-14-00784-f003:**
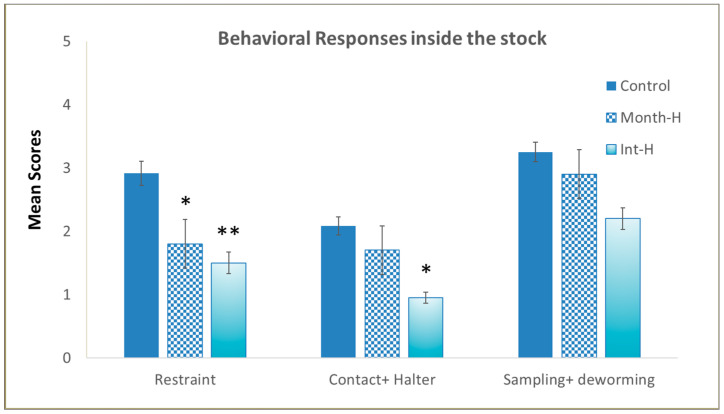
Stock test. Ranking scores, ranging from 0 to 5, with 0 being the lowest and 5 the maximum reactivity responses to being isolated, restrained, haltered, blood sampled, and dewormed in the different experimental groups of foals (N = 9–12 per group) (mean values ± S.E.). * *p* < 0.05 and ** *p* < 0.01 indicate significant differences compared with controls.

**Table 1 animals-14-00784-t001:** Human- and test-related responses observed during experimental behavioral tests.

	Behavioral Response	Score	Description
Responses to environments	a. Isolation and restraint; b. Contact and halter; c. Blood sampling and deworming.	0	No movements of feet.
		1	Some calm movements of feet (less than 10).
		2	Continuous movements of feet, head up.
		3	One or two abrupt movements with avoidance behaviors (head up, rear up).
		4	More than two sudden movements with defensive behaviors (back toward handler)
		5	Danger to itself and others (e.g., falling back, putting head or limbs on bars)
Responses toward humans	a. Quality of contact	0	Does not allow
		1	Neck touched
		2	*Plus* shoulder
		3	*Plus* back
		4	*Plus* limbs
		5	All over the body
	b. Evasive behaviors	0	Stays calm and do not move while the person approaches
		1	Moves, walking
		2	Trot
		3	Canter
		4	Jump/gallop

Description of each score assigned to the animal’s behaviors adapted from Pereira-Figueiredo [31].

**Table 2 animals-14-00784-t002:** Behaviors toward humans, during approach–contact test.

	Behavioral Item	Description
Positive behaviors	Attention to the person [27]	Neck high, ears, and head forward toward person
	Exploration of the person [35]	Neck horizontal or lower, sniffing and ears forward, toward person
	Approaching the person	Turning itself and moving toward handler
Negative behaviors	Defensive behaviors [6]	Turning back, ears back

Behaviors are described as the total duration of each behavior in seconds throughout the duration of the approach–contact test (180 s).

**Table 3 animals-14-00784-t003:** Time taken and ranking scores in forced-person test in the foals from the different experimental groups not handled (controls N = 10) or handled (Int-H, N = 11, and Month-H, N = 9), during MHT and approach–contact test (mean values ± S.E.).

	Group	Control	Int-H	Month-H	*p*-Value
MHT 60 s	Rx isolation	30.4 ± 3.1	26.2 ± 3.0	25.4 ± 3.6	ns
	Lat zone (in s)	58.8 ± 5.1	55.2 ± 3.5	59.4 ± 3.7	ns
	D Apo (in m)	1.02 ± 0.5	0.82 ± 0.14	0.97 ± 0.3	ns
	Lat touch (in s)	143.2 ± 23.6	115.5 ± 32.7	121.3 ± 25.5	ns
	Quality contact	8.9 ± 2.8	19.8 ± 4.1 *	12.9 ± 3.2	0.031
Approach–contact test 180 s	Evasive behaviors	48.3 ± 4.2	21.4 ± 3.9 *	22.3 ± 4.4 *	0.015
	Positive behaviors (in s)	30.2 ± 5.1	9.5 ± 4.2 *	11.0 ± 4.5	0.039
	Negative behaviors (in s)	37.8 ± 5.2	16.5 ± 3.8 *	18.0 ± 3.9 *	0.028

* *p* < 0.05 indicates significant differences compared with controls. ns indicates no significance.

**Table 4 animals-14-00784-t004:** Plasma and serum values obtained in the arterial blood for all foals (controls, Month-H, and Int-H) during the stock test. Mean values ± S.E. in the different experimental groups. Abbreviations: LDH: lactate dehydrogenase; CK: creatine kinase; WBC: leukocyte cell count.

	Biochemical Counts			Hematological	Counts (×10^9^/L)	
	CK	LDH	WBC	Neutrophils	Lymphocytes	NL Ratio
Reference Values	21–473	0–1830	6–12	0.6–4.0	2.5–7.5	0.8–2.2
Control (N = 10)	201.1 ± 14.8	1704.4 ± 78.8	10.9 ± 5.3	5.79 ± 0.74	3.99 ± 0.31	1.46 ± 0.21
Month-H (N = 8)	268.8 ± 42	1736.0 ± 48.5	11.26 ± 4.9	7.65 ± 0.45	3.45 ± 0.37	2.41 ± 0.26
Int-H (N = 11)	225.3 ± 14.2	1676.8 ± 99.8	11.07 ± 5.2	6.94 ± 0.63	3.12 ± 0.38	2.55 ± 0.24 *

* *p* < 0.05 indicates significant differences compared with controls. The color red indicates values over the reference values for equines. Reference intervals were established by ADVIA Hematology Analyzer (Bayer, Leverkusen, Germany).

## Data Availability

Data are contained within the article.

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
