# Peer review of "Forced Handling Decreases Emotionality but Does Not Improve Young Horses’ Responses toward Humans and their Adaptability to Stress"

_animals, 2024, doi:10.3390/ani14050784_

Round 1
Reviewer 1 Report
Comments and Suggestions for Authors
The scientific article titled "Forced Handling does not improve young horses’ relationship to humans and adaptability to stress" by Inês Pereira-Figueiredo, Ilda Rosa, and Consuelo Sancho. The article discusses the influence of different approaches of human contact in young horses and their later adaptability to humans and environments. The study involved 31 foals separated into three groups: one submitted to gentle handling, one to "negative" handling, and one left undisturbed. The foals were evaluated during behavioral tests and physiological assessments at 8 months old. The article discusses the potential impact of early handling on the behavior, welfare, and adaptability of young horses, as well as the implications for their relationship with humans and their ability to face new and stressful situations.
The article highlights several important aspects related to the study:
Weaknesses and Limitations: The study focuses on a specific set of handling procedures and their effects on young horses. It does not consider a wide range of handling methods or other potential factors that could influence the outcomes. The sample size is relatively small, and the study only involves Lusitano foals from a specific location, which may limit the generalizability of the results.
Testability of the Hypothesis: The hypothesis that early handling, if gentle, can be a positive experience and improve the behavioral outcomes of young horses, their welfare, and their ability to face new and stressful situations is testable through the behavioral tests and physiological assessments conducted on the foals.
Methodological Inaccuracies: The study's methodology involves behavioral observations, physiological assessments, and handling procedures. However, the specific details of the physiological assessments and the behavioral observation methods are not fully described in the provided excerpt.
Economic Implications: The article does not explicitly discuss economic implications. However, the findings of the study could have implications for the training and handling practices of young horses, which could in turn impact the economic aspects of horse breeding and training.
Literature Review: The article provides a comprehensive review of the existing literature on the topic, discussing the importance of positive human-animal relationships, the potential impact of early handling on young horses, and the existing knowledge gaps in this area.
Implications: The study's findings suggest that early forced handling is stressful and does not improve adaptability to humans and new stressors in young horses. This has implications for the development of handling practices that promote positive human-animal relationships and the well-being of young horses.
Overall, the article provides valuable insights into the potential effects of different handling approaches on young horses' behavior and adaptability, highlighting the need for further research and the development of evidence-based handling practices in the equine industry.
Introduction:
Lines 47-49: In my opinion, the opening sentence should be more impactful to directly state the focus on horses at the start.
Lines 50-51: Consider clarifying the term "umwelt" as it may not be familiar to all readers. Forexample you can add an sentence: “This concept, known as 'umwelt', is particularly relevant in the context of horses, which humans use for sports, leisure, and even therapies."
Lines 52-54: The transition between caring for horses and the prevalence of poor conditions is a bit abrupt. It might help to add a connecting sentence for smoother flow.
Lines 55-59: This section can be improved, could be more concise. Consider combining sentences to streamline the information.
Lines 60-64: Please consider in this section here to simplify for clarity.
Lines 65-71: In this paragraph here, I would suggest you to consider emphasizing the novelty of the positive human-animal relationship in welfare studies earlier in the paragraph. A lot of studies have been conducted in this sense 10.3390/ani12141740)
Lines 72-77: In this part here the transition to the early handling of young horses is a bit abrupt. You should consider to introduce this topic with a sentence that connects it to the previous discussion about human-animal relationships (10.1186/s12917-022-03289-2)
Material and methods:
General comment: Please in this part here you should insert the written part in which you clarify the ethical consideration for the study methods and protocol. It is very important to safeguard the animal welfare.
Lines 81-82: In my opinion you should consider adding a rationale for why Lusitano foals were specifically chosen for this study.
Lines 94-101: It would be beneficial to add more detail about how these conditions replicate or differ from the foals' natural environment, to contextualize the study's relevance.
Lines 109-121: In this part here, I would recommend you to clarify the rationale for choosing the specific handling procedures and their relevance to the study's objectives.
Lines 211-233: In my opinion, it would be beneficial to explain why these specific parameters were chosen and how they relate to the study's aims.
Results
The results are well-written, supporting the study objectives
Please consider only minor details such as the to remove the space, verify and adapt for the rest of manuscript.
Discussions and conclusion
The discussions are comprehensive, well-written and well-structured. However, you should consider to add a few more connections between the findings and the broader implications for horse welfare and handling practices would make it more impactful. Here are some of my specific comments:
Lines 319-333: I would suggest you in this part here to briefly mention how the results align with or differ from the initial hypothesis. For example, did the study find that good handling reduces fearfulness in young horses as hypothesized?
Lines 334-369: In this section here, consider adding a concluding sentence to each paragraph that ties the findings back to the central question of the study. For instance, how do these findings contribute to the understanding of handling's impact on young horses?
Lines 370-401: The discussion on the behavioural responses of horses towards humans is detailed and well-explained. It might be helpful to briefly summarize the key findings at the end of this section for clarity.
Lines 402-426: I suggest you to include a rationle on the implications of these findings for horse welfare. For example, what do the elevated neutrophil-lymphocyte ratios in Gentle-H foals suggest about their stress levels?
Lines 427-436: The conclusion effectively summarizes the study's findings. You can enhance your conclusions by including recommendations for future research or practical applications of the study.
For example, how might these findings influence horse training or welfare practices?
Author Response
Dear reviewer, we are very thankful for your positive observations about our manuscript, we appreciated your suggestions and tried to change it accordingly. All changes in the re-submitted manuscript are highlighted at gray.
Dear reviewer, our study only involves Lusitano foals from a specific location, as in this research we aimed to evaluate the conditions that such breed farms can have in later fearfulness and relationship to humans. This is explained now in Introduction, from line 80 until line 90: “In the South of Portugal and Spain, is very common to young horses be reared on large breeding farms and kept together in big groups of the same age on pasture. These social conditions”…
The specificity of the breed is justified later from Line 105: “…However, some studies have led to contradictory results, possibly due to heterogeneity of study designs, the moment handling was performed, the type, or duration of the contact. Furthermore, research on this topic quite often uses horses from different breeds, different housing conditions, with ignorance of the history of previous contacts. Regarding the high influence of individual differences in temperament traits and behavioral responses towards humans [34- 37], controlling such variables can be of upmost importance. To our knowledge, our study is the first one testing behavioral responses in young horses raised under the same conditions, age, and breed, and this can provide a great opportunity for a better understanding of human interactions and handling practices during horses’ development
The specificity of handling procedure is explained at the beginning of 2.2 Handling procedure subtopic from Line 179: The Handling procedure was adapted from our previous work [31], and followed the methodology first described by Miller [46] and modified by Henry [28] and Schmideck [29], regarding the moment it was performed and being done with the foals standing.
As suggested we included a new subtopic about the limitations of the study in Discussion: 4.4 Limitations of the study and concerns about handling practices. And from Lines 544 to 553 we included a paragraph: “Overall, our results must be taken cautiously as they had some limitations. Non-independence of study participants may have affected groups behavior and hematologic findings - future research to test external validity of findings with an external group of horses is needed. Other important limitations arise from the small sample sizes and the specificity of handling procedures used in our study, as we recognize that small sample sizes can in-crease the risk of false positive findings. The present study makes part of a big project that aims to test the effects of different time frames of handling in later responses of foals, therefore it did not consider a wide range of handling methods or other potential factors that could influence the outcomes, and these may limit the generalizability of the results..”….
Methodological Inaccuracies: We appreciated your comments very much, therefore we included 3 paragraphs with details of the physiological assessments and the behavioral observation methods in introduction, from Lines 112-133: “Currently, there is a growing interest of biomarkers of good or bad welfare. The most common physiological indicators of stress and discomfort, in several species, include cortisol levels (blood and saliva), muscle tension, or heart rate variability (HRV) [20], [38]. Nevertheless, until now, there is still a poor conformity on the reliability and validity of such measures of horse’s emotional states. The HR response to experimental situations varies a lot among individuals, and HR increases with emotional reactivity is not always displayed behaviorally [19]. Cortisol is often not the best indicator of stress and poor welfare in horses, as its secretion increases with physical activity, follows a daily fluctuation [39], is influenced by gender, and may be decreased in those that experience altered welfare [17].
Regardless of the inherent connection among behavioral and physiological responses to stress, this is not thoroughly studied. Ethograms usually comprise objective and instantaneous indicators of welfare status in animals [16], [40], but it is still difficult to measure emotional states in horses [21], and most of emotional behaviors are only revealed under high anxiety conditions [31]. In a previous study from our lab, it was shown that hematological (hematocrit, platelet, and leukocyte counts) and biochemical parameters (protein and cytoplasmic enzyme total counts) were altered in rodents exposed to prolonged stress, even months after it had ended [41], suggesting that these markers could be good indicators of chronic stress. Also, better than leukocyte counts, the ratio of Neutrophil:lymphocyte (N:L ratio) was suggested as a reliable biomarker measuring stress in vertebrates [42,43]. And recently, some authors proposed the assessment of N:L ratio as a good indicator of stress and welfare status in horses [16], [44].
Economic Implications are included in a new paragraph from lines 66 to 77: “Any of the described conditions is of huge importance for horses’ welfare and may set off serious health problems [4], that can shorten their working life, and bring economic costs to their trainers or owners. This may result in immunosuppression [12], learning issues [5], [11], including lack of motivation [9], [13], etc., and can be the cause of many accidents, mostly because of highly disproportional emotional responses.”
Introduction:
1 Lines 52-54: the opening sentence is more impactful and directly focusing on horses : Humans and horses, for the last centuries, have been having a common history, with a close relationship, establishing strong bonds [1-2]. But not always horses were fully understood and not always they understood human activities. This paradigm can be explained if comparing equines evolution as a prey animal, we remember their domestication is quite recent.
2; 3; 4 and 5: Thank you for your comments. They have been very helpful; we changed the rationale of the introduction and now introduction is more based on the results of the study. We kept only this phrase about umwelt “Horses have been evolving for millions of years as a sensitive animal, very perceptive of their surrounding world” We combined some sentences and tried to reorganize it to better explain the purpose of the research. We included a final sentence in the paragraphs explaining the impact of each point, as in Line 64…”horses may become easily frightened and stressed.
6 and 7: We included a paragraph emphasizing the novelty of the positive human-animal relationship in welfare studies from Line 91: “In this perspective, recent studies converge on the idea that promoting positive interactions with humans, can be a way to improve horses’ emotionality, social bonds, and well-being [20-25]. It is argued that human company may be important to a horse, where physical contact, and the benefits from stroking, are crucial elements to potentiate human attachment [23-26]. Likewise, the frequency and number of interactions can determine how horses deal with situations imposed by humans. Revising the literature suggests that human interactions may simply consist of habituation to any contact stimulus (brushing, haltering, etc.) [21-22], [25] and its continuous repetition, until becoming neutral, has been named handling [26-28].”
Material and methods:
To better clarify the ethical consideration for the study methods and protocol we included the sentence: The authors confirmed that animal welfare was never at risk and that the experiments complied with the policy relating to animal ethics and welfare (Article 1, number 5, Directive 2010/63/EU of the European Parliament and of the Council of 22 September 2010 on the protection of animals used for scientific purposes; and Article 2, number 7, of the Portuguese Legislation number 113/2013 of August 2013).
We included in material and methods from Line 142: “ Considering the genetic component linked to emotionality and temperament traits we evaluated Lusitanian foals only [45]…..
The conditions of breed and environment during the study replicate the foals' natural conditions Line 144-145: “As usual practice of the Stud farm, until weaning, mares, and foals (a total of 57 born from the end of winter and spring of 2021), were kept on pasture during the night (from 4 pm until 8 am), being housed every morning.”
As explained in Lines 178-181: The Handling procedure was adapted from our previous work [31], and followed the methodology first described by Miller [46] and modified by Henry [28] and Schmideck [29], regarding the moment it was performed and being done with the foals standing.
In introduction we included 2 paragraphs( from Lines 112-134), explaining about the blood parameters chosen and how they relate to the study's aims.
Discussions
Thank you for your comments: We added a few more connections between the findings and the broader implications for horse welfare and handling practices
We added “According to previous research, the results obtained showed that early handling, regardless of its intensity, resulted in a learned behavior, with a decreased self-defense and emotional responses towards humans and stressful situations [25, 26], [30, 31].
In subtopic: 4.1. Handling decreased emotional reactivity, regardless of its intensity.
“With handling, fundamentally, the animal learns by habituation [27] to identify irrelevant specific stimuli and diminishes reactivity towards it [12]. This is a feature of extreme im-portance in equines, due to the combination of a large and highly reactive animal, in which sudden defensive responses to pain or fear may easily injure handlers, veterinarians, or trainers [48,49]. In the present study, it was observed that handling decreased later reactivity or emotional responses of the young horses, to get restrained in a stock, and to be touched and haltered by one unknown person, with no differences in response to blood sampling and deworming. None of the handled foals achieved the highest reactivity response (score 5) during routine veterinary procedure or harmed themselves (comparing to 3 from Control group).”
We added a concluding sentence to each paragraph:
In subtopic: 4.1. Handling decreased emotional reactivity, regardless of its intensity.
“In the present study, it was observed that handling decreased later reactivity or emotional responses of the young horses, to get restrained in a stock, and to be touched and haltered by one unknown person, with no differences in response to blood sampling and deworming …..Notwithstanding, as demonstrated in this assay, more information is needed, regarding the emotional states of the horse, whether positive, or altered, and its responses towards humans”.
In subtopic 4.2. Handling decreased negative but also positive behaviors towards humans
……. Our results may suggest that handling, daily or monthly, decreased emotional responses towards humans but did not improve bonding will.”
We included some paragraphs about the implications of our findings for horse welfare from Line 496: “…We could also establish that lymphopenia was not explicit and differential WBC count was normal. These results agree with previous works [7], asserting even more the importance of the indicative value of this ratio (N:L). In a recent study, Popescu [16] found increased values of the N:L ratio in working horses and suggested that these values were probably consequence from inadequate housing and management, after investigating the relationship between the N:L ratio and welfare scores.
In the present study, the handled foals showed mean values of the N:L ratio just a trace above reference values, but this means that there were some individuals for which the N:L ratio were higher. Moreover, comparing the range of N:L ratios, we observed that 70% of Control foals were within the normal range for healthy horses (1.25~1.57) [44], but only 10% (Int-H) and 12% (Month-H) of handled ones were also within this range, and all the others exhibited higher scores.
Altogether, we cannot conclude that our findings indicate a health problem in the young horses- as all other parameters were at normal values (after examining weights, biochemical and parasitological status)- nor to issues in the working program, housing, or management, as all foals were raised together in the same conditions. Implicating that the elevated N:L ratios we found, rather suggest discomfort and low welfare, in previous handled foals.
and conclusion
At conclusion we included recommendations for future research or practical applications of the study.

Reviewer 2 Report
Comments and Suggestions for Authors
Unfortunately, it was very difficult to review this paper due to the errors in the English language. As a result, I do not feel that I was able to provide an in-depth review. However, here are some comments for authors that could be addressed further after the English language of the paper is corrected.
1. In the introduction, add some information and references about your chosen methods of determining stress and welfare, especially the use of neutrophil-lymphocyte ratio as a measure of animal welfare. How quickly does this ratio change when an animal is stressed?
2. In order to compare the "gentle" handling technique to the "negative" handling technique, why was the timing of the handling sessions so different? Why did the researchers perform all "gentle" handling procedures in 3 days, but performed the "negative" handling over 3 months? Why not match the "gentle" technique to the "negative" technique by also performing the "gentle" technique over 3 months?
3. I would like more explanation of the method for ranking behaviors described in line 242-245. How did the authors choose to rank behaviors and why?
4. I am confused by the conclusions of this paper. The conclusion stated in the title "Forced handling does not improve young horses' relationship to humans and adaptability to stress", seems to be incongruous with many of the results such as:
a. Line 251: " Handling significantly affected the foals' responses inside a stalk, with handled ones displaying lower reactivity when isolated and restrained"
b. Line 255: "we found that Gentle-H foals tolerated better human contact than Controls"
c. Line 268: "Control foals were more difficult to reach and displayed more defensive reactions towards an unknown handler, when compared to handled ones"
d. line 278: " Gentle-H foals allowing more body areas to be touched than Controls"
I understand that the researchers also noted that handled foals exhibited fewer "positive" behaviors towards humans on tests and some changes on blood tests, but I think there is also some important information disregarded here (see examples above).
Comments on the Quality of English Language
Unfortunately, it was very difficult to review this paper due to the errors in the English language. I found most of the procedure descriptions difficult to follow and understand, and as a result am having a hard time providing constructive feedback for the authors. I would suggest having an English translator who is familiar with this type of research to go over this manuscript with the authors before resubmitting this paper.
Author Response
Dear Reviewer: Thank you very much for taking the time to review this manuscript. Please find the detailed responses below and the corresponding revisions and corrections highlighted in gray in the re-submitted file.
- We appreciated your comments very much, and we included 3 paragraphs with details of the hematological assessments and the behavioral observation methods chosen of determining stress and welfare. In introduction, from Lines 112-133: “Currently, there is a growing interest of biomarkers of good or bad welfare. The most common physiological indicators of stress and discomfort, in several species, include cortisol levels (blood and saliva), muscle tension, or heart rate variability (HRV) [20], [38]. Nevertheless, until now, there is still a poor conformity on the reliability and validity of such measures of horse’s emotional states. The HR response to experimental situations varies a lot among individuals, and HR increases with emotional reactivity is not always displayed behaviorally [19]. Cortisol is often not the best indicator of stress and poor wel-fare in horses, as its secretion increases with physical activity, follows a daily fluctuation [39], is influenced by gender, and may be decreased in those that experience altered welfare [17]. Regardless of the inherent connection among behavioral and physiological responses to stress, this is not thoroughly studied. Ethograms usually comprise objective and instantaneous indicators of welfare status in animals [16], [40], but it is still difficult to measure emotional states in horses [21], and most of emotional behaviors are only revealed under high anxiety conditions [31]. In a previous study from our lab, it was shown that hematological (hematocrit, platelet, and leukocyte counts) and biochemical parameters (protein and cytoplasmic enzyme total counts) were altered in rodents exposed to prolonged stress, even months after it had ended [41], suggesting that these markers could be good indicators of chronic stress. Also, better than leukocyte counts, the ratio of Neutrophil:lymphocyte (N:L ratio) was suggested as a reliable biomarker measuring stress in vertebrates [42,43]. And recently, some authors proposed the assessment of N:L ratio as a good indicator of stress and welfare status in horses [16], [44]."
We included more information in Discussion about the time taken to change Neutrophil counts in peripheral blood:
Lines 488-490: "The significant increase on the Neutrophil counts in peripheral blood has been reported in several species and just after 6 min of a stressful event, exercise, or inflammation of the muscle tissues [53].”
2 We explained in Introduction, lines 136-139: "By comparing outcomes in different approaches of handling procedure (more or less intense), the present study hopes to contribute to a better understanding of the impact human that activities have on equines development of later behaviors, welfare, and quality of life."
In Discussion we changed the sentence from Line 395: ”The present study aimed to define if short term handling experiences in young horses would reduce fearfulness and improve later responses toward humans and new and stressful situations. To do so, foals submitted to different intensities of handling procedures (daily or monthly sessions) were compared.
We changed the terms “Gentle handling” and “Negative handling” to intensive (Int-H) and monthly handling (Month-H), as the intensity of handling was the main difference among the interventions. As for example in Material and Methods, Lines 152- 154: “One group was submitted to daily sessions, intensive handling Int-H), another was submitted to once monthly sessions (at 3, 4 and 5 months old: monthly Handling: Month-H) and the last group was left undisturbed (Controls).
- The description of each Score assigned to the animal’s behaviors was adapted from our previous study Pereira-Figueiredo et al. (2017) http://dx.doi.org/10.1016/j.applanim.2017.06.016) that itself was an adaptation from Ligout et al. (2008);Popescu and Diugan ( 2013) Forkman et al. (2007).
4. To better recognize the results of our study we changed the title “Forced Handling decreases emotionality but does not improve young horses’ responses towards humans and adaptability to stress
In Introduction, we included several sentences regarding the importance of increasing safety for handlers and the animal itself:
From line 66…. “Any of the described conditions is of huge importance for horses’ welfare and may set off serious health problems [4], that can shorten their working life, and bring economic costs to their trainers or owners. This may result in immunosuppression [12], learning issues [5], [11], including lack of motivation [9], [13], etc., and can be the cause of many accidents, mostly because of highly disproportional emotional responses.”
Emotional behavior, reactivity, or emotionality is defined as the “set of behavioral and physiological responses that occur in anxiety producing circumstances” [14] and how to control it is a main factor in equines industry. Identify any strategy that would help them to reduce fearfulness, and modulate their behavioral responses to situations and humans, when facing potential stressful-conditions, is an essential component impacting not only their health but horse-related incidents and accidents with their handlers.
At Discussion we included, from Line 395: “The present study aimed to define if short term handling experiences in young horses would reduce fearfulness and improve later responses toward humans and new and stressful situations.
From Lines 413: With handling, fundamentally, the animal learns by habituation [27] to identify irrelevant specific stimuli and diminishes reactivity towards it [12]. This is a feature of extreme im-portance in equines, due to the combination of a large and highly reactive animal, in which sudden defensive responses to pain or fear may easily injure handlers, veterinari-ans, or trainers [48,49].
In the present study, it was observed that handling decreased later reactivity or emotional responses of the young horses, to get restrained in a stock, and to be touched and haltered by one unknown person, with no differences in response to blood sampling and deworming. None of the handled foals achieved the highest reactivity response (score 5) during routine veterinary procedure or harmed themselves (comparing to 3 from Control group).

Reviewer 3 Report
Comments and Suggestions for Authors
See file uploaded

Comments on the Quality of English LanguageEnglish Language Comments
Throughout the manuscript, it is evident that English is not the authors’ first language. Overall, however, the research purpose, methods, and findings are clear, despite grammatical issues related to English language conventions and grammar.
Some of the phrasing is unusual or incorrect, but the meaning is clear.
e.g., ‘state of spirit’ in lines 331-333. “Blood and fecal samples were 331 collected, for hematological and parasitological analysis, and for determination of any interference in animal welfare and therefore in their state of spirit.”
e.g., the use of restrain (verb)/restraint (noun), e.g., Lines 340-341. “In the present study, handling foals, gently or not, improved behavioural responses to restrain and to later veterinary procedures,…”.
e.g., incorrect use of commas throughout the text.
In several instances the word selection or grammatical format could create confusion.
e.g., throughout the article the authors use the term ‘stalk’ but are probably referring to stocks.
e.g., addition of the word ‘yet’, Lines 420-421. “The cytosolic enzymes in blood samples (eg. CK and LDH parameters) in our study yet, demonstrated no changes among conditions and experimental groups,
The article needs a very close review on grammar and style before publication.
e.g., Frequent use of the word ‘despite’, e.g., Line 206. “The approach did not stop if the foal moved walking, and despite contact was not forced…”, Line 277. “Despite the Quality of contact was significantly affected with previous handlings 277 (χ2(2) =11.5, p=0.031)”; Line 298. “Despite the leukocyte and lymphocyte 298 profiles were not affected,…”.
In text citation formatting is incorrect
The authors frequently use the citation number in place of the authors’ names to complete a sentence. The author(s) should be named, not substituted with a citation number. It is distracting as a reader and occurs throughout the paper. Here are some selected examples.
· Line 347: “When [12] compared different methods…”
· Line 113: “…procedure adapted from [17]”.
· Line 119: “It excluded some of the desensitization actions used at [17] study, such as touching the mouth….”
· Lines 389-392. “Despite [15] observed that gently handled foals did not discriminate handlers, regarding their familiarity or experience, [13] suggested the need to consider some human attributes, while studying human-animal interactions and welfare.”
Author Response
Dear Reviewer:
Thank you very much for taking the time to review this manuscript. Please find the detailed responses below and the corresponding revisions and corrections highlighted in gray in the new submited manuscript.
We are very thankful for your positive comments on our manuscript. We took all of it into great attention and tried to follow all recommendations. We sent the article to be reviewed on grammar and style by our translation team from the University of Salamanca. We started by checking the correct use of the word restraint, instead of restrain, we changed stalk to stock, and we corrected text citation formatting, including the authors names.
GENERAL COMMENTS
1 regarding your concerns we changed the title to Forced Handling decreases emotionality but does not improve young horses’ responses towards humans and adaptability to stress. As you suggest we changed the terms “Gentle handling” and “Negative handling” to intensive (Int-H) and monthly handling (Month-H), as the intensity of handling was the main difference among the interventions.
2 Thank you for your comments. They have been very helpful, we changed the rationale of the introduction and this is now more based on the results of the study.
In Introduction we included several sentences and paragraphs, we took into consideration and better explained the impact of the brief handling experiences and being reared in high-output breeding farms: From line 80 until line 90: “In the South of Portugal and Spain, is very common to young horses be reared on large breeding farms and kept together in big groups of the same age on pasture. These social conditions are potentially positive for equines, when comparing to individually housed ones [16-18]. Horses kept in group learn quicker, decrease undesirable behaviors, and require less desensitization time to novel equipment [18], [19]. Nevertheless, in such high-breeding farms, fillies and colts are “artificially” put together, by humans’ decision, very different from the natural composition of social groups [6]. Furthermore, in these breeding farms, the interactions of the young animals with humans are normally based in short occasional veterinary inspections and besides feeding, positive contacts are very rare. Altogether, horses kept on growing in such breeding conditions can develop potential difficult relationships with peers and humans….” In Discussion we included some sentences regarding the effects of brief handling experiences
3 Our study only involves Lusitano foals from a specific location, as in this research we aimed to evaluate the conditions that such breed farms can have in later fearfulness and relationship to humans. This is explained now in Introduction, from line 80:… ”In the South of Portugal and Spain, is very common to young horses be reared on large breeding farms and kept together in big groups of the same age on pasture. These social conditions”…
The specificity of the breed is justified later from Line 105: “…However, some studies have led to contradictory results, possibly due to heterogeneity of study designs, the moment handling was performed, the type, or duration of the contact. Furthermore, research on this topic quite often uses horses from different breeds, different housing conditions, with ignorance of the history of previous contacts. Regarding the high influence of individual differences in temperament traits and behavioral responses towards humans [34- 37], controlling such variables can be of upmost importance. To our knowledge, our study is the first one testing behavioral responses in young horses raised under the same conditions, age, and breed, and this can provide a great opportunity for a better understanding of human interactions and handling practices during horses’ development,
We included in material and methods from Line 143: “ Considering the genetic component linked to emotionality and temperament traits we evaluated Lusitanian foals only [45]…..
Moreover, we included a new subtopic about the limitations of the study in Discussion: 4.4 Limitations of the study and concerns about handling practices. And from Lines 544 to 553 we included a paragraph: “Overall, our results must be taken cautiously as they had some limitations. Non-independence of study participants may have affected groups behavior and hematologic findings - future research to test external validity of findings with an external group of horses is needed. Other important limitations arise from the small sample sizes and the specificity of handling procedures used in our study, as we recognize that small sample sizes can in-crease the risk of false positive findings. The present study makes part of a big project that aims to test the effects of different time frames of handling in later responses of foals, therefore it did not consider a wide range of handling methods or other potential factors that could influence the outcomes, and these may limit the generalizability of the results..”….
Specific Comments
- The 1st sentence in the Abstract “Inappropriate stressful conditions may lead to several health consequences” was replaced to a more specific one: “Horses are often still exposed to stressful or inadequate conditions and difficult relationships with humans, despite growing concerns about animal welfare”.
5 we replaced the term stalk to stock.
6 In Introduction, we included several sentences regarding the importance of increasing safety for handlers and the animal itself:
…. “Any of the described conditions is of huge importance for horses’ welfare and may set off serious health problems [4], that can shorten their working life, and bring economic costs to their trainers or owners. This may result in immunosuppression [12], learning issues [5], [11], including lack of motivation [9], [13], etc., and can be the cause of many accidents, mostly because of highly disproportional emotional responses.”
Emotional behavior, reactivity, or emotionality is defined as the “set of behavioral and physiological responses that occur in anxiety producing circumstances” [14] and how to control it is a main factor in equines industry. Identify any strategy that would help them to reduce fearfulness, and modulate their behavioral responses to situations and humans, when facing potential stressful-conditions, is an essential component impacting not only their health but horse-related incidents and accidents with their handlers.
At Discussion we included From Line 395: “The present study aimed to define if short term handling experiences in young horses would reduce fearfulness and improve later responses toward humans and new and stressful situations.
From Lines 413: With handling, fundamentally, the animal learns by habituation [27] to identify irrelevant specific stimuli and diminishes reactivity towards it [12]. This is a feature of extreme im-portance in equines, due to the combination of a large and highly reactive animal, in which sudden defensive responses to pain or fear may easily injure handlers, veterinarians, or trainers [48,49].
In the present study, it was observed that handling decreased later reactivity or emotional responses of the young horses, to get restrained in a stock, and to be touched and haltered by one unknown person, with no differences in response to blood sampling and deworming. None of the handled foals achieved the highest reactivity response (score 5) during routine veterinary procedure or harmed themselves (comparing to 3 from Control group). ….
7 Lines 52-56: the opening sentence is more impactful and directly focusing on horses: Humans and horses, for the last centuries, have been having a common history, with a close relationship, establishing strong bonds [1-2]. But not always horses were fully understood and not always they understood human activities. This paradigm can be explained if comparing equines evolution as a prey animal, we remember their domestication is quite recent.
8 Thank you for your comments. We kept only this phrase about umwelt “Horses have been evolving for millions of years as a sensitive animal, very perceptive of their surrounding world and we included several paragraphs, eg. emphasizing the novelty of the positive human-animal relationship in welfare studies from Line 91: “In this perspective, recent studies converge on the idea that promoting positive interactions with humans, can be a way to improve horses’ emotionality, social bonds, and well-being [20-25]. It is argued that human company may be important to a horse, where physical contact, and the benefits from stroking, are crucial elements to potentiate human attachment [23-26]. Likewise, the frequency and number of interactions can determine how horses deal with situations imposed by humans. Revising the literature suggests that human interactions may simply consist of habituation to any contact stimulus (brushing, haltering, etc.) [21-22], [25] and its continuous repetition, until becoming neutral, has been named handling [26-28].”
9 As suggested and above explained we changed the terms positive and negative and included several sentences about handling procedure: In Introduction, From Line 91: “In this perspective, recent studies converge on the idea that promoting positive inter-actions with humans, can be a way to improve horses’ emotionality, social bonds, and well-being [20-25]. It is argued that human company may be important to a horse, where physical contact, and the benefits from stroking, are crucial elements to potentiate human attachment [23-26]. Likewise, the frequency and number of interactions can determine how horses deal with situations imposed by humans. Revising the literature suggests that human interactions may simply consist of habituation to any contact stimulus (brushing, haltering, etc.) [21-22], [25] and its continuous repetition, until becoming neutral, has been named handling [26-28].
In Material and Methods, From Line 178: "The Handling procedure was adapted from our previous work [31], and followed the methodology first described by Miller [46] and modified by Henry [28] and Schmideck [29], regarding the moment it was performed and being done with the foals standing"
And in discussion, From Line 413: "With handling, fundamentally, the animal learns by habituation [27] to identify irrelevant specific stimuli and diminishes reactivity towards it [12]. This is a feature of extreme im-portance in equines, due to the combination of a large and highly reactive animal, in which sudden defensive responses to pain or fear may easily injure handlers, veterinarians, or trainers [48,49]."
10 we included in flow chart caption information about blood sampling. … “After weaning, all experimental foals (N=31) were subjected to two consecutive days of Behavioural tests, including isolation, restraint, blood Sampling and deworming
Also in the section 2.3.1 Stock Test, there are information about the procedure (lines 236 and 237).
11 a) We changed Table 1 and included a Table 2
- b) Negative and positive Behaviors were examined as total time and were not scored. As it is in the Caption: Description of each Score assigned to the animal’s behaviours or
- c) We included examples of score 5 and changed the sentence of loss of self-esteem to Danger to self and others (eg. to fall back, putting head or members at bars)
12 We changed to Motionless Human test (MHT) Line 261.
13 we included 3 paragraphs with details of the hematological assessments and the behavioral observation methods chosen of determining stress and welfare. In introduction, from Lines 112-133: “Currently, there is a growing interest of biomarkers of good or bad welfare. The most common physiological indicators of stress and discomfort, in several species, include cortisol levels (blood and saliva), muscle tension, or heart rate variability (HRV) [20], [38]. Nevertheless, until now, there is still a poor conformity on the reliability and validity of such measures of horse’s emotional states. The HR response to experimental situations varies a lot among individuals, and HR increases with emotional reactivity is not always displayed behaviorally [19]. Cortisol is often not the best indicator of stress and poor welfare in horses, as its secretion increases with physical activity, follows a daily fluctuation [39], is influenced by gender, and may be decreased in those that experience altered welfare [17].
Regardless of the inherent connection among behavioral and physiological responses to stress, this is not thoroughly studied. Ethograms usually comprise objective and instantaneous indicators of welfare status in animals [16], [40], but it is still difficult to measure emotional states in horses [21], and most of emotional behaviors are only revealed under high anxiety conditions [31]. In a previous study from our lab, it was shown that hematological (hematocrit, platelet, and leukocyte counts) and biochemical parameters (protein and cytoplasmic enzyme total counts) were altered in rodents exposed to prolonged stress, even months after it had ended [41], suggesting that these markers could be good indicators of chronic stress. Also, better than leukocyte counts, the ratio of Neutrophil:lymphocyte (N:L ratio) was suggested as a reliable biomarker measuring stress in vertebrates [42,43]. And recently, some authors proposed the assessment of N:L ratio as a good indicator of stress and welfare status in horses [16], [44].
14 High parasite fecal counts can affect health status ant thence behavioral responses · Therefore, for this study it was important to make sure all foals were healthy and behavioral responses were not affected by health issues also we could also have information to discussion From Line 509: ” we cannot conclude that our findings indicate a health problem in the young horses- as all other parameters were at normal values (after examining weights, biochemical and parasitological status)
15 We are very thankful for your input about the static model you suggest us to include. We had it in great consideration for this and for future manuscripts (like one we are almost finishing).
16 we specified the sample size at Table 2
17 Positive behaviors were descripted as total duration (total time behavior lasted in seconds through the duration of test: 180 s). For better clearness we included a new table (Table 3) where each descriptor is defined.
Therefore we tried to better explain this in Lines 351 to 357: When the total time exhibiting positive behaviors towards the unknown person was assessed, it was determined that handling also affected the will to positively interact with the handler (F1,31= 12.8 p=0.039), and post-hoc analyses showed that Int-H foals exhibited significantly fewer positive interactions than Controls (9.5 vs. 30.2, p=0.028), with the same trend in Monthly handled ones (p=0.052).
18 Significant differences at N:L Ratio were found, as an effect of handling (χ2(2) =21.5, p=0.047). Levels of N:L Ratio of previously handled foals were over reference values and post-hoc revealed that Int-H foals presented higher levels than Controls (2.55 vs. 1.46, p=0.042), with no differences in Month-H foals.
19 We corrected the Figure 3 to Stock test; included X-units; and included in caption information about the score meaning.
20
- a) We changed to N:L ratio references as this is the correct term. b) We corrected Table 3; and clarified the meaning of red font at caption.
- d) In discussion some paragraphs have been included from Line 488: “The significant increase on the Neutrophil count in peripheral blood has been reported in several species and just after 6 min of a stressful event, exercise, or inflammation of the muscle tissues [53]. In this study, it could be demonstrated that cytosolic enzymes in blood (eg. CK and LDH parameters) were not altered in any experimental group, meaning the Neutrophilia was not linked to muscle injury, or inflammatory state, and most probably was linked to the stress condition.
The increased number and percentage of neutrophils accomplished with lymphopenia is becoming a recommended tool for the physiological assessment of distress in ani-mals, for a growing number of researchers [42-44]. We could also establish that lympho-penia was not explicit and differential WBC count was normal. These results agree with previous works [7], asserting even more the importance of the indicative value of this ratio (N:L). In a recent study, Popescu [16] found increased values of the N:L ratio in working horses and suggested that these values were probably consequence from inadequate housing and management, after investigating the relationship between the N:L ratio and welfare scores.
21 Some sentences were included in discussion regarding our findings and what literature suggests. We included some paragraphs about the implications of our findings for horse welfare from Line 496: “…We could also establish that lymphopenia was not explicit and differential WBC count was normal. These results agree with previous works [7], asserting even more the importance of the indicative value of this ratio (N:L). In a recent study, Popescu [16] found increased values of the N:L ratio in working horses and suggested that these values were probably consequence from inadequate housing and management, after investigating the relationship between the N:L ratio and welfare scores. In the present study, the handled foals showed mean values of the N:L ratio just a trace above reference values, but this means that there were some individuals for which the N:L ratio were higher. Moreover, comparing the range of N:L ratios, we observed that 70% of Control foals were within the normal range for healthy horses (1.25~1.57) [44], but only 10% (Int-H) and 12% (Month-H) of handled ones were also within this range, and all the others exhibited higher scores. Altogether, we cannot conclude that our findings indicate a health problem in the young horses- as all other parameters were at normal values (after examining weights, biochemical and parasitological status)- nor to issues in the working program, housing, or management, as all foals were raised together in the same conditions. Implicating that the elevated N:L ratios we found, rather suggest discomfort and poor welfare, in previous handled foals."
22 as you suggested we included in the introduction (already above in 9) and changed the terms good handling and gentling experiences for handling and forced handling is more discussed.
23 We have changed it.
24 instead of attachment we emphasize the fact that the foals only contacted with humans for a very short time frame, but tried to explain that these can be important in animals breed with very few contact to humans: Lines 86-90: "Furthermore, in these breeding farms, the interactions of the young animals with humans are normally based in short occasional veterinary inspections and besides feeding, positive contacts are very rare. Altogether, horses kept on growing in such breeding conditions can develop potential difficult relationships with peers and humans"
25 We altered “changes in behavioral and physiological responses to hematological parameters".
26 we included in Line 453: "However, more research is needed on the extent of generalization of handling experiences".
27 we included a new subtopic: 4.4 Limitations of the study and concerns about handling practices where some of your concerns are included. From Line 545 “Overall, our results must be taken cautiously as they had some limitations. Non-independence of study participants may have affected groups behavior and hematologic findings - future research to test external validity of findings with an external group of horses is needed. Other important limitations arise from the small sample sizes and the specificity of handling procedures used in our study, as we recognize that small sample sizes can increase the risk of false positive findings. The present study makes part of a big project that aims to test the effects of different time frames of handling in later responses of foals, therefore it did not consider a wide range of handling methods or other potential factors that could influence the outcomes, and these may limit the generalizability of the results. Furthermore, some correlations between health-related welfare indicators and behavioral responses of the assessed horses as follow ups would be helpful. Still, it must be noted that the patterns of behavioral and hematological changes here described are of upmost importance to consider in further research”.

Round 2
Reviewer 1 Report
Comments and Suggestions for Authors
the paper improved a lot